# Establishment of a Rat Model of Alcoholic Liver Fibrosis with Simulated Human Drinking Patterns and Low-Dose Chemical Stimulation

**DOI:** 10.3390/biom13091293

**Published:** 2023-08-24

**Authors:** Chien-Yu Lin, Evanthia Omoscharka, Yanli Liu, Kun Cheng

**Affiliations:** 1Division of Pharmacology and Pharmaceutical Sciences, School of Pharmacy, University of Missouri-Kansas City, 2464 Charlotte Street, Kansas City, MO 64108, USA; 2Department of Pathology, University Health/Truman Medical Center, School of Medicine, University of Missouri-Kansas City, 2301 Holmes Street, Kansas City, MO 64108, USA

**Keywords:** alcoholic liver fibrosis, alcohol liquid diet, binge, CCl_4_, rat model

## Abstract

Although alcohol is a well-known causal factor associated with liver diseases, challenges remain in inducing liver fibrosis in experimental rodent models. These challenges include rodents’ natural aversion to high concentrations of alcohol, rapid alcohol metabolism, the need for a prolonged duration of alcohol administration, and technical difficulties. Therefore, it is crucial to establish an experimental model that can replicate the features of alcoholic liver fibrosis. The objective of this study was to develop a feasible rat model of alcoholic liver fibrosis that emulates human drinking patterns and combines low-dose chemicals within a relatively short time frame. We successfully developed an 8-week rat model of alcoholic liver fibrosis that mimics chronic and heavy drinking patterns. Rats were fed with a control liquid diet, an alcohol liquid diet, or alcohol liquid diet combined with multiple binges via oral gavage. To accelerate the progression of alcoholic liver fibrosis, we introduced low-dose carbon tetrachloride (CCl_4_) through intraperitoneal injection. This model allows researchers to efficiently evaluate potential therapeutics in preclinical studies of alcoholic liver fibrosis within a reasonable time frame.

## 1. Introduction

Alcohol abuse is a significant health problem that contributes to numerous metabolic and liver diseases [1]. The liver, as a major organ for alcohol metabolism, plays a crucial role in converting alcohol to acetaldehyde via alcohol dehydrogenase and cytochrome P450 2E1 (CYP2E1), which is further oxidized to acetate by acetaldehyde dehydrogenase [2]. Alcohol use disorder causes severe liver injury and inflammation, ultimately resulting in the development of steatohepatitis (>90%), liver fibrosis (20–40%), cirrhosis (8–20%), and hepatocellular carcinoma (HCC) (3–10%) [3,4]. Chronic consumption of alcohol not only increases the permeability of endotoxins in the intestines but also increases the levels of oxidative stress and the production of acetaldehyde and cytokines, all of which cause liver damage [5,6]. In addition, acute alcohol intake, also known as binge drinking, can induce acute alcoholic hepatitis and accelerate the progression of liver pathogenesis in individuals with chronic alcoholism [7].

Rodent models of liver fibrosis can be generated by a variety of etiologic factors. Among them, chemical-induced liver fibrosis models are widely used because of their high reproducibility [8]. Particularly, carbon tetrachloride (CCl_4_) is the most commonly employed hepatotoxin to study liver fibrosis and cirrhosis in rodents. However, the CCl_4_-induced liver fibrosis model is not able to fully replicate the pathology and mechanism of alcohol-induced liver fibrosis [9]. Although alcohol is a well-known etiologic factor and exhibits a synergistic effect with other risk factors in the development of liver diseases [10,11], there is currently no rodent model that can easily perform and accurately recapitulate all of the features of alcohol-induced liver fibrosis [12]. The establishment of an animal model for alcoholic liver fibrosis is a technically challenging, labor-intensive, and time-consuming process. Maintaining high blood alcohol levels in rodents is difficult compared to humans, primarily due to their inadequate intake of alcohol and their high rate of alcohol metabolism [12]. Gradually increasing the amount of alcohol in drinking water (A-DW) is the simplest way to administer alcohol and mimic human drinking behavior. Although the A-DW model can initiate steatosis, it requires additional stimulators to induce further inflammation and fibrotic development [13]. The Tsukamoto–French model is considered the most effective approach for inducing liver fibrosis and administering large amounts of alcohol to maintain high blood alcohol levels in mice [14]. However, this model requires surgical implantation of intragastric alcohol administration, which has a higher mortality rate and is not a true physiological model [15].

Alternatively, the Lieber–DeCarli liquid diet model is widely used for long-term chronic alcohol consumption in rodents. This model forces rodents to consume an isocaloric alcohol-containing liquid diet as their only source of food and drink [15,16]. Although it is not a perfect physiological model, administering the Lieber–DeCarli liquid diet is a milder approach that induces mild steatosis, low-grade inflammation, but no fibrosis in the liver. This is suitable for studying the early stages of alcoholic liver diseases [16]. To simulate advanced liver pathogenesis, Dr. Bin Gao’s group developed a modified version of the Lieber–DeCarli liquid diet, known as the National Institute of Alcohol Abuse and Alcoholism (NIAAA) model. In contrast to the chronic alcohol feeding or binge alone models, the NIAAA model combines chronic Lieber–DeCarli liquid diet feeding and multiple binge alcohol sessions. This model closely resembles the clinical features of human alcoholic steatohepatitis, including elevated serum levels of alanine transaminase (ALT), aspartate transaminase (AST), blood alcohol levels, and neutrophil infiltration [15,17].

Rats are genetically and physiologically closer to humans than mice. This makes rat a preferred animal model for biomedical research, especially studies related to gene modulation. Therefore, it is critical to develop a highly reproducible rat model that can recapitulate alcohol-associated liver diseases in humans. Previous reports have suggested that combining an alcohol liquid diet or a western diet and secondary hepatotoxic substances, such as CCl_4_ and diethylnitrosamine (DEN), can induce the progression of liver fibrosis and the development of HCC [9,16,18,19,20]. In this study, we established a rapid and feasible rat model of alcoholic liver fibrosis that simulates patterns of chronic and heavy drinking observed in humans. Considering alcohol as the primary etiologic factor, we decreased the dose and frequency of CCl_4_ administration compared to the conventional CCl_4_-induced liver fibrosis model [8]. The addition of low-dose CCl_4_ (0.1 mL/kg) significantly enhances the fibrogenic effect of alcohol. By combining alcohol and low-dose CCl_4_, we were able to rapidly induce a consistent and reproducible liver fibrosis model within an 8-week timeframe.

## 2. Materials and Methods

### 2.1. Ethics Statement

The alcoholic liver fibrosis model was conducted using Sprague Dawley rats, which were purchased from the Jackson Laboratory and housed in a temperature- and humidity-controlled room on a 12 h light/dark cycle. The animal research protocol (protocol number: 1110) was approved by the University of Missouri-Kansas City (UMKC) Institutional Animal Care and Use Committee (IACUC), and all experiments conformed to the relevant regulatory standards.

### 2.2. Dietary and CCl_4_ Treatment

Eight-week-old male Sprague Dawley rats were used to establish the rat model of alcoholic liver fibrosis. The diets and administration schedule of alcohol binge and CCl_4_ are illustrated in Figure 1A. The rats were randomly assigned to the following groups: control liquid diet, control liquid diet with CCl_4_ administration four times (once weekly, QW), alcohol liquid diet, alcohol liquid diet with CCl_4_ administration four times (QW), alcohol liquid diet with CCl_4_ administration four times (once every 2 weeks, Q2W), alcohol liquid diet plus binge, alcohol liquid diet plus binge with CCl_4_ administration twice (Q2W), and alcohol liquid diet plus binge with CCl_4_ administration four times (Q2W). Each group consisted of 7–9 rats. The Lieber–DeCarli 82 control liquid diet (F1259) and alcohol liquid diet (F1258) were prepared according to the manufacturer’s protocol (Bio-Serv, Frenchtown, NJ). To mimic human chronic alcohol consumption and binge drinking patterns, the rats were administered the alcohol liquid diet containing 5% alcohol on a daily basis. Additionally, alcohol binges (5 g alcohol per kg of body weight) were given twice every week (TIW). A mixture of CCl_4_ and olive oil (1:4) was intraperitoneally (i.p.) injected at a dose of CCl_4_ 0.1 mL/kg. The injections were given once a week for 4 weeks (QW) or once every 2 weeks for 8 weeks (Q2W). All rats were euthanized after 8 weeks. Blood samples were collected for serological analysis, and the livers were harvested for histological examination, cytokine analysis, and mRNA expression analysis.

### 2.3. Serum Analysis

Whole blood samples were collected using heparinized tubes. These tubes were centrifuged for 10 min at 1500× *g* and 4 °C to separate the plasma. The plasma samples were sent to the Clinical Pathology Laboratory at the University of Missouri for analysis of various parameters, including glucose, urea nitrogen, creatinine, albumin, total protein, globulin, cholesterol, total bilirubin, ALT, AST, alkaline phosphatase (ALP), and glutamate dehydrogenase (GLDH).

### 2.4. Liver Histology

Liver biopsy specimens were fixed with formalin, embedded in paraffin, and sectioned into 5 µm thick slices. The tissue slides were stained with hematoxylin and eosin (H&E) for histopathological assessment and with Sirius Red for fibrosis assessment. The Sirius Red-positive areas were quantified using ImageJ software. To evaluate the degree of fibrosis in the liver specimens, fibrosis scores were assigned based on the Ishak stage score ranging from 0 to 6, as previously reported [21].

### 2.5. Hydroxyproline Assay

The hepatic collagen content was measured using the hydroxyproline assay as previously reported [22]. Briefly, 50 mg of liver tissue was homogenized in 250 μL of PBS. Then, 250 μL of the tissue homogenate was transferred to a glass vial containing 500 µL of 12 N HCl and incubated overnight at 120 °C in a dry bath incubator. The acid hydrolyzed homogenate was filtered using a 0.45 polyvinylidene fluoride (PVDF) filter. Then, 20 µL of sample and hydroxyproline standards were added to a 96-well plate and mixed thoroughly with 100 μL of Chloramine T solution at room temperature for 30 min. Finally, 100 μL of Ehrlich’s solution (Sigma-Aldrich, St. Louis, MO, USA) was added to each well and incubated at 65 °C for 15–20 min. The photometric product was determined using a microplate reader at wavelength 550 nm. The amount of hydroxyproline (µg) was calculated relative to the liver weight (g).

### 2.6. Immunohistochemistry

The formalin-fixed paraffin-embedded liver sections were deparaffinized and incubated in a citric acid antigen retrieval solution (10 mM citric acid, 0.05% Tween 20, pH 6) for 45 min at 95–100 °C to expose the epitope. The presence of specific markers on liver sections was evaluated according to the protocol of the immunohistochemistry kit (ab64261, Abcam, Chou City, Tokyo). The liver sections were incubated with primary antibodies against 𝛼-smooth muscle actin (𝛼-SMA) (#19245, Cell Signaling Technology, Danvers, MA, USA), suppressor of mothers against decapentaplegic 2/3 (Smad2/3) (#8685, Cell Signaling Technology), poly (rC) binding protein 2 (Pcbp2) (GWB-3815A, GenWay Biotech), and Ki-67 (#D3B5, Cell Signaling Technology) overnight at 4 °C. All of the antibodies were diluted as suggested by the manufacturer’s instructions.

### 2.7. Enzyme-Linked Immunosorbent Assay (ELISA)

Liver tissues weighing approximately 100–200 mg were homogenized in RIPA Lysis and Extraction Buffer containing Protease Inhibitor Cocktail (Thermo Scientific, Rockford, IL, USA). The levels of various cytokines and chemokines in the liver tissues, including interleukin-1β (IL-1β), IL-4, IL-6, IL-10, IL-18, transformation growth factor-β (TGF-β), tumor necrosis factor-α (TNF-α), interferon-γ (IFN-γ), monocyte chemoattractant protein-1 (MCP-1), regulated upon activation, normal T cell expressed and secreted (RANTES) (R&D Systems Inc., Minneapolis, MN, USA), and programmed death-ligand 1 (PD-L1) (MyBioSource, San Diego, CA, USA), were measured by corresponding commercial ELISA kits according to the manufacturer’s protocol.

### 2.8. Quantitative Real-Time Polymerase Chain Reaction (PCR)

Approximately 50–100 mg of liver tissue was homogenized in 1 mL TRIzol™ Reagent (Invitrogen, Carlsbad, CA, USA), and the total RNA was isolated and purified according to the manufacturer’s protocol. The extracted RNA samples were analyzed with a real-time PCR detection system (Bio-Rad, Hercules, CA, USA) using the iTaq™ Universal SYBR^®^ Green One-Step Kit according to the manufacturer’s protocol. The expression of type I collagen (*Col1a1*) gene at mRNA levels was determined using the comparative cycle threshold method with *18S* as the reference [23].

### 2.9. Statistical Analysis

The statistical analysis was performed using Excel and GraphPad Prism 6.0 software. All data are presented as mean ± SEM. The difference in significance was determined using one-way analysis of variance (ANOVA), followed by Dunnett’s multiple comparison test. *p* < 0.05 was considered statistically significant.

## 3. Results

### 3.1. General Characteristics of the Animal Models

Body weight was recorded to assess the impact of types of different diets, alcohol binge, and low-dose CCl_4_ treatment on the biological conditions of the rats. At the end of the study, rats fed with alcohol liquid diet (c–h) exhibited significantly lower weight gain compared to the control diet feeding groups (a,b) (Figure 1B). Furthermore, the administration of alcohol binge twice a week further decreased weight growth (f–h) compared to rats fed with alcohol diet alone (c–e).

While low-dose CCl_4_ had little effect on weight gain in the control diet feeding group (b), a significant body weight loss was observed after low-dose CCl_4_ was given in the alcohol diet feeding groups (d,e,g,h). Rats treated with low-dose CCl_4_ at once weekly and once every 2 weeks (d,e) did not show differences in body weight gain at the end of the study. As expected, increasing the frequency of CCl_4_ administrations resulted in reduced body weight gain (g,h). The fluctuation in body weight reflected the combined effect of alcohol binge and CCl_4_ administration, which significantly reduced the activity and food intake of the rats. Additionally, the toxicity of alcohol feeding and the combined chemical toxin was assessed using the liver-to-body weight ratio [24]. The alcohol diet feeding groups did not exhibit increased liver weight compared to the control diet feeding groups (Figure 1C). However, the groups that received alcohol diet plus binge and CCl_4_ demonstrated the highest liver weight to body weight ratio (Figure 1D).

### 3.2. Plasma Analysis for Liver Injury and Metabolism

The metabolic profiles of the animals are presented in Figure 2. There were no significant differences in blood glucose levels (Figure 2A), urea nitrogen (Figure 2B), creatinine (Figure 2C), total bilirubin (Figure 2D), and ALP (Figure 2J) among all groups. However, in rats treated with the alcohol diet plus binge and CCl_4_ × 4 (Q2W), albumin and total protein levels showed significant decreases (Figure 2E,F), indicating the presence of chronic conditions that affected the liver or kidney. Rats fed with the alcohol diet plus binge exhibited a significant decrease in plasma levels of globulin (Figure 2G). Cholesterol levels were significantly increased in rats treated with the alcohol diet plus CCl_4_ × 4 (QW) (Figure 2H), while no statistical differences were observed in the remaining groups treated with the alcohol diet. Liver enzyme activities are commonly used as biomarkers for liver damage. In this study, ALT was significantly increased in rats fed with alcohol diet, particularly in rats treated with low-dose CCl_4_ (Figure 2I), which is consistent with the features of the alcohol liquid diet plus binge/secondary hit models [25]. Interestingly, we did not observe elevated AST levels in any of the treated groups (Figure 2K). This could be because AST is not specific for the liver because the heart has the highest concentration of AST. Moreover, AST is present in various other tissues including muscles, kidneys, brain, pancreas, and erythrocytes [26]. Another potential reason is that ALT and AST levels primarily indicate acute liver injury. Consequently, their levels in cases of chronic liver injury are lower than those observed in acute liver injury. For example, Gao et al. observed that in chronic liver injury induced by chronic alcohol feeding plus multiple binges of alcohol, the serum ALT and AST levels were lower when compared to chronic alcohol consumption combined with a single binge [17].

On the other hand, GLDH is widely employed as a specific alternative biomarker for liver injury. This mitochondrial enzyme is mainly located within the liver lobule and is released into the blood from damaged hepatocytes [27,28]. GLDH has shown better sensitivity and specificity compared to ALT in the detection of liver injuries. Compared to rats fed with the control diet, plasma levels of GLDH were significantly increased in rats fed with alcohol diets (Figure 2L), which is consistent with hepatocyte necrosis observed in patients with alcoholic liver diseases [29,30].

### 3.3. Liver Histology and Liver Fibrosis Assessment for the Animal Models

The histological characteristics of representative livers from each group are shown in Figure 3. In the control diet group (a), normal cellular integrity was observed. However, small-droplet and large-droplet fat were found in rats fed with alcohol diet (c–e) and alcohol diet plus binge (f–h). Although the two control diet groups (a,b) showed no differences in general features, exposure to low-dose CCl_4_ could induce the accumulation of lipid droplets in the liver. Specifically, rats treated with a combination of alcohol and a low-dose CCl_4_ were more susceptible to fat accumulation, which resulted in mild to moderate steatosis. Additionally, lobular inflammation was found in rats fed with alcohol diet.

Sirius Red staining is a useful method for detecting collagen in liver tissue sections. Figure 4A,B demonstrate the Sirius Red staining of representative rats and quantification of the Sirius Red-positive area for each group. Table 1 summarizes the fibrosis stage of each group based on the Ishak stage score (0–6) [31]. In rat groups fed with control liquid diet (a,b), collagen was expressed at normal levels, and no fibrosis or short fibrosis septa was observed in rats receiving additional low-dose CCl_4_ administration. By contrast, the expression levels of collagen significantly increased in the alcohol diet groups and alcohol diet groups with additional low-dose CCl_4_ treatment (d, e, g, and h). Fibrosis expansion and bridging fibrosis developed in all rats treated with the combination of alcohol and CCl_4_, but not in rats treated with alcohol alone. Moreover, the expression levels of *Col1a1* mRNA were elevated in all rats fed with alcohol liquid diet and subjected to four doses of CCl_4_ treatment (d, e, and h), reaching eight-fold higher than the control liquid diet group (Figure 4C). The hydroxyproline content in the liver consistently showed increased collagen accumulation in rats treated with the combination of alcohol and CCl_4_ (Figure 4D).

### 3.4. Cytokine and Chemokine Assessments for Alcohol and CCl_4_ Treated Animal Models

Cytokines, namely IL-6, IL-10, and TNF-𝛼, are strongly associated with the development of alcoholic liver diseases according to previous research [6]. Compared to the control diet group, the levels of IL-6 declined slightly in rats fed with alcohol diet (Figure 5A). The addition of CCl_4_ further decreased the levels of IL-6, suggesting a reduced protective effect of IL-6 during the early stage of alcohol-induced liver diseases. IL-10, an anti-inflammatory cytokine, was remarkably decreased in rats treated with an alcohol diet plus binge and CCl_4_ administration every 2 weeks (Q2W) (Figure 5B). In general, levels of the systemic inflammatory cytokine TNF-𝛼 are known to be elevated in the serum during the progression of both nonalcoholic steatohepatitis and alcoholic liver diseases [32,33]. However, no statistically significant changes were observed in the liver (Figure 5C). Furthermore, pro-inflammatory cytokines (IL-18, IL-4, IFN-γ, and IL-1β) as well as chemokines (MCP-1 and RANTES) did not show significant elevation in the liver (Figure 5D–I).

### 3.5. Synergistic Effect of Alcohol and CCl_4_ on Activation of Hepatic Stellate Cells

The activation of hepatic stellate cells (HSCs) is the primary factor that facilitates the progression of liver fibrosis, playing a critical role in the production and accumulation of extracellular matrix (ECM) [34]. Figure 6 shows the immunostaining and quantification of markers for activated HSCs in the liver tissue sections. Rats fed with the control diet (a), alcohol diet (c), and alcohol diet plus binge (f) feeding alone did not increase the α-SMA staining area (Figure 6A,D). However, all rats fed with the alcohol diet and four administrations of low-dose CCl_4_ (d,e,h) showed a broad 𝛼-SMA-positive area, indicating that HSCs can be efficiently activated by the combination of alcohol and low-dose CCl_4_ treatment within 8 weeks. In addition, the upregulation of the TGF-β/Smad signaling pathway indicated the progression of liver fibrosis (Figure 6B,E,G). In accordance with the previous report, alcohol and cytokines can upregulate the mRNA levels of *PCBP2* in primary rat HSCs [35]. The immunostaining results demonstrated that PCBP2 was upregulated in all rats treated with either alcohol or low-dose CCl_4_ alone, and the combination of alcohol and low-dose CCl_4_ promoted the higher expression levels of PCBP2 (Figure 6C,F).

### 3.6. Effect of Alcohol and CCl_4_ on Cell Proliferation and PD-L1 Expression

The nuclear protein Ki-67 is commonly used as a marker for identifying cell proliferation. Based on the immunostaining results, the number of Ki-67-positive nuclei was significantly increased with the combination of alcohol and low-dose CCl_4_ stimulation compared to the control diet group (a), especially in rats treated with four doses of CCl_4_ (d,e,h) (Figure 7A,B). The increased abundance of hepatocyte and non-parenchymal cell proliferation indicates chronic liver inflammation and fibrosis and serves as a potential indicator for the development of hepatic carcinogenesis [36]. Although the levels of PD-L1 expression in rats treated with alcohol/CCl_4_ showed a slight suppression compared to the control diet group, there were no statistical differences in all groups (Figure 7C).

## 4. Discussion

In this study, we established easily operated rat models for alcoholic liver fibrosis and evaluated the fibrotic features of the designed models. Rats were administered a chronic alcohol liquid diet and multiple binges via oral gavage to mimic chronic and heavy alcohol consumption in humans. There were no significant differences between rats fed with an alcohol diet and rats fed with an alcohol diet plus multiple binges in terms of the development of liver fibrosis. This finding is consistent with previous reports [15]. Dr. Gao’s group suggested that multiple binges are considered as chronic alcohol administration for mice fed with an alcohol diet, which showed lower elevations of serum AST and ALT compared to mice treated with alcohol feeding plus a single binge [17]. Moreover, the technical challenges associated with oral gavage, such as the risk of the gavage needle and liquid entering the trachea, could increase the mortality rate.

Steatosis is the most common histological appearance of liver diseases caused by alcohol intake in the initial stage. The consumption and metabolism of alcohol contribute to the upregulation of lipogenic enzymes, which leads to increased synthesis of triglyceride and phospholipid, resulting in excessive lipid accumulation [37,38]. On the other hand, CCl_4_ is a potent hepatotoxin that is metabolized by CYP450 enzymes in the liver. Its free radical metabolites, trichloromethyl radical (CCl_3_^●^) and trichloromethylperoxy radicals (CCl_3_OO^●^), exhibit high reactivity with proteins, nucleic acids, and lipids, leading to fat accumulation and the advanced pathogenic development in the liver [39,40]. The histological structure of our alcoholic liver fibrosis models, induced by alcohol and low-dose CCl_4_, reproduced the characteristic features of lipid accumulation and lobular inflammation observed in patients with alcoholic liver diseases [41].

To increase reproducibility and accelerate the progression of alcoholic liver fibrosis, the addition of a secondary hepatotoxin such as CCl_4_ may be an optimal approach. Several studies have utilized chronic alcohol feeding along with CCl_4_ to induce liver injury and fibrosis, as well as to evaluate the hepatoprotective effect of potential therapeutics in rodent models [9,42,43,44,45,46,47,48,49,50,51,52]. However, CCl_4_, as a secondary hit or accelerating agent, does not directly correspond to human disease. CCl_4_ is highly impacted by other hepatotoxins and diets, which may not perfectly resemble the features of human alcoholic liver fibrosis [53]. Therefore, we aimed to minimize the impact of CCl_4_ on the initiation of liver fibrosis by using four doses of low-dose of CCl4 (0.1 mL/kg). By comparison, 8–12 i.p. injections of CCl_4_ (1 mL/kg) are needed to establish liver fibrosis in rats without alcohol feeding. CCl_4_ can be administered through either vapor inhalation [44,45] or i.p. injection [46,47], but i.p. injection is more convenient and consistent. While rodent models co-administered with alcohol and CCl_4_ have been employed in previous research, only a very limited number of studies have examined the pathophysiological patterns of these combined treatments.

Our models of alcohol and CCl_4_ co-administration revealed that the interval of low-dose CCl_4_ treatment did not show remarkable differences in the development of liver fibrosis. By contrast, the number of low-dose CCl_4_ treatments had a significant impact on the progression of liver fibrosis (Figure 1). The combination of alcohol and low-dose CCl_4_ exhibited a synergistic effect, leading to accelerated liver damage, steatosis, inflammation, and fibrosis. As expected, markers of liver injury and fibrosis, including liver enzyme activities, accumulation of lipid droplets, expression of collagen, activation of HSCs, and cell proliferation, were significantly elevated following stimulation with alcohol and low-dose CCl_4_. Notably, it was observed that the fibrotic levels were consistently similar among rats within each group, indicating a high level of reproducibility in the animal models. The increased mRNA level of the *Col1a1* gene was found to be associated with the induction of PCBP2 expression by alcohol. PCBP2 is an RNA binding protein that can stabilize the *Col1a1* mRNA, thereby contributing to ECM accumulation in the liver [35]. In addition, a study suggested that PCPB2 overexpression was found in HCC patients with poor prognosis, indicating its potential as a prognostic marker for HCC [54]. Overall, the combination of alcohol and four doses of low-dose CCl_4_ i.p. injection effectively enhanced the progression of liver fibrosis in rats within 8 weeks, thus providing a reliable alcoholic liver fibrosis model for future studies.

In addition to HSCs, chronic alcohol exposure has significant effects on other liver cells. This includes inducing organelle stress in hepatocytes, altering the structure of hepatic sinusoidal endothelial cells, and influencing the population and functions of immune cells [55,56]. Natural killer (NK) cells are believed to play an important role in eliminating activated HSCs, but alcohol consumption can reduce the number of NK cells [57]. Moreover, alcohol intake enhances the activity of Kupffer cells, which is closely associated with fibrogenesis in the liver. A study demonstrated that combined treatment with alcohol and CCl_4_ in a rat model increased the number of Kupffer cells [50]. Following alcohol and CCl_4_ treatment, the activated Kupffer cells upregulated the production of profibrotic factors such as TGF-β and inflammatory cytokines [56,58].

Our rats treated with alcohol and CCl_4_ exhibited a significant increase in cell proliferation. The expression level of Ki-67, a marker for cell proliferation, is associated with the degree of inflammation and the stage of fibrosis, but it does not increase proportionally [36]. Ki-67 positive hepatocytes and bile ductal cells were significantly suppressed in the end-stage of alcoholic cirrhotic liver [59]. The programmed cell death 1 (PD-1)/PD-L1 pathway plays a critical role in maintaining immune tolerance [60]. PD-L1 is associated with protective immunity, but its expression is suppressed during liver damage [61,62]. Studies have demonstrated abundant expression of PD-L1 on Kupffer cells and liver sinusoidal epithelial cells [63,64]. In patients with chronic hepatitis B, elevated expression of PD-L1 and PD-L2 was observed compared to non-viral hepatitis cases, where no upregulation of PD-L1 and PD-L2 was found [63]. Furthermore, PD-L1 overexpression has been observed in HCC tumors and surrounding tissues, and it is correlated with tumor aggressiveness and overall survival rate [63,65,66,67].

In summary, we have developed rat models for alcoholic liver fibrosis by utilizing a combination of hepatotoxic agents, alcohol, and low-dose CCl_4_. The inclusion of a low-dose CCl_4_ proved to be highly effective in accelerating the progression of liver fibrosis in rats fed with alcohol within an 8 week time frame. Among all models, the combination of an alcohol liquid diet and binge drinking with low-dose CCl_4_ administration four times (model h, Figure 1A) showed the best effect in inducing liver injury and fibrosis. On the other hand, using an alcohol liquid diet with low-dose CCl_4_ administration four times (models d and e, Figure 1A) could be an easier and alternative procedure to induce a similar degree of chronic liver injury and fibrosis. These models provide novel experimental platforms for the study of pathological mechanisms and drug screening in the context of alcoholic liver fibrosis.

## Figures and Tables

**Figure 1 biomolecules-13-01293-f001:**
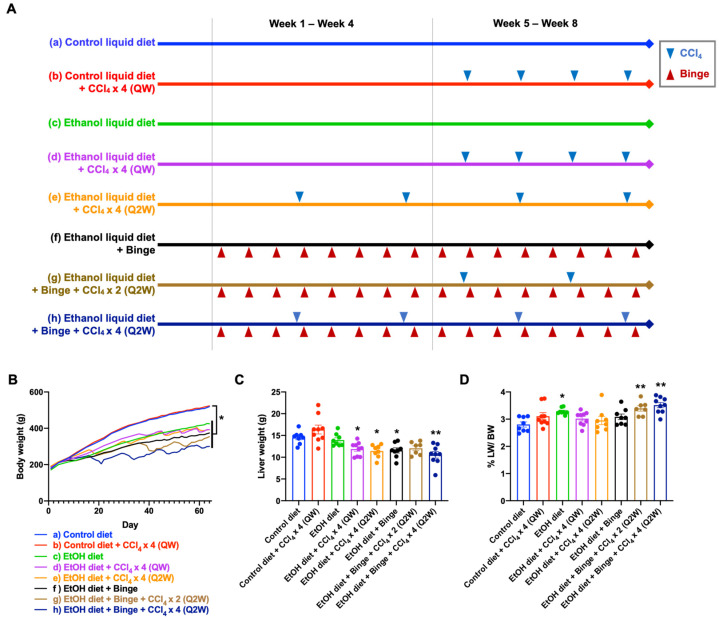
Procedures for animal models and general characteristics. (**A**) The types of diets and schedules of ethanol binge and CCl_4_ administrations: (a) control liquid diet; (b) control liquid diet and four doses of CCl_4_ (QW); (c) alcohol liquid diet; (d) alcohol liquid diet and four doses of CCl_4_ (QW); (e) alcohol liquid diet and four doses of CCl_4_ (Q2W); (f) alcohol liquid diet plus binge; (g) alcohol liquid diet plus binge and two doses of CCl_4_ (Q2W); (h) alcohol liquid diet plus binge and four doses of CCl_4_ (Q2W). (**B**) Body weight. (**C**) Liver weight and (**D**) liver-to-body weight ratio at the endpoint of the study. Results are presented as the mean ± SEM (n = 7–9; * *p* < 0.05; ** *p* < 0.01).

**Figure 2 biomolecules-13-01293-f002:**
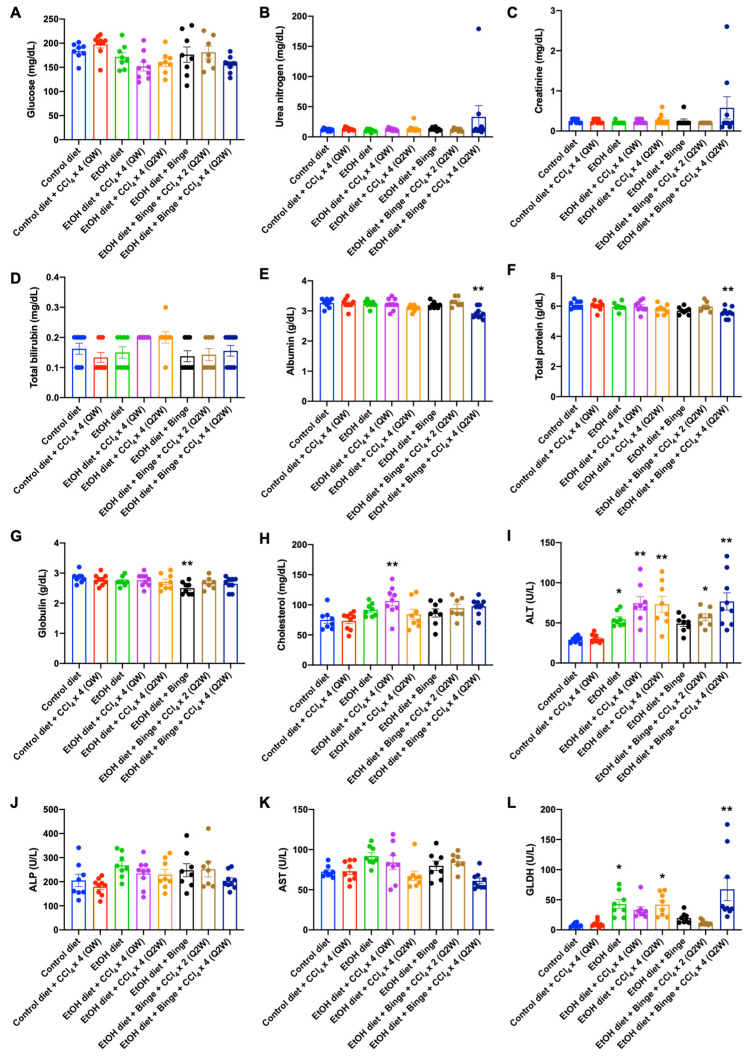
Metabolic profiles of the animals in each model. Plasma levels of (**A**) glucose, (**B**) urea nitrogen, (**C**) creatinine, (**D**) total bilirubin, (**E**) albumin, (**F**) total protein, (**G**) globulin, (**H**) cholesterol, (**I**) ALT, (**J**) ALP, (**K**) AST, and (**L**) GLDH. Results are presented as the mean ± SEM (n = 7–9; * *p* < 0.05; ** *p* < 0.01).

**Figure 3 biomolecules-13-01293-f003:**
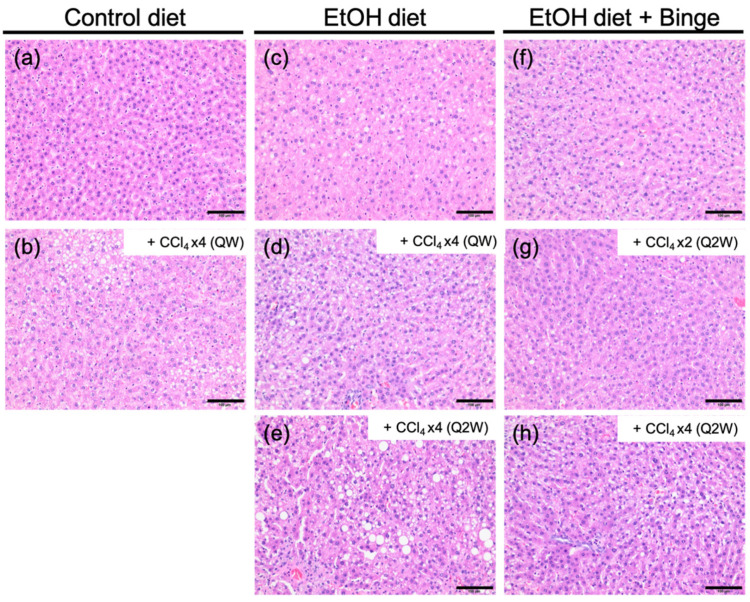
Histological features of liver tissues. H&E staining of representative rats treated with (**a**) control liquid diet; (**b**) control liquid diet and four doses of CCl_4_ (QW); (**c**) alcohol liquid diet; (**d**) alcohol liquid diet and four doses of CCl_4_ (QW); (**e**) alcohol liquid diet and four doses of CCl_4_ (Q2W); (**f**) alcohol liquid diet plus binge; (**g**) alcohol liquid diet plus binge and two doses of CCl_4_ (Q2W); (**h**) alcohol liquid diet plus binge and four doses of CCl_4_ (Q2W). The scale bar represents 100 µm.

**Figure 4 biomolecules-13-01293-f004:**
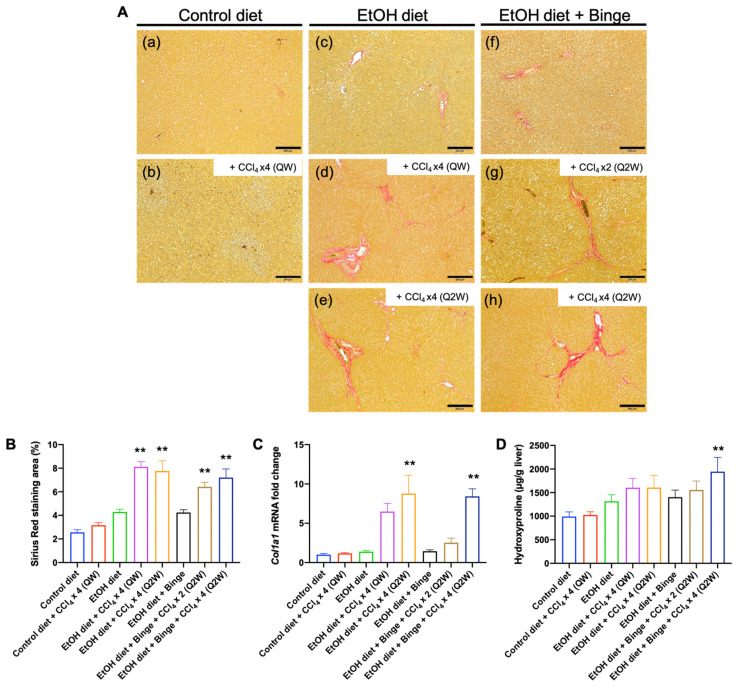
Liver fibrosis assessment of the animal models. (**A**) Sirius Red staining of representative rats treated with (**a**) control liquid diet; (**b**) control liquid diet and four doses of CCl_4_ (QW); (**c**) alcohol liquid diet; (**d**) alcohol liquid diet and four doses of CCl_4_ (QW); (**e**) alcohol liquid diet and four doses of CCl_4_ (Q2W); (**f**) alcohol liquid diet plus binge; (**g**) alcohol liquid diet plus binge and two doses of CCl_4_ (Q2W); (**h**) alcohol liquid diet plus binge and four doses of CCl_4_ (Q2W). The scale bar represents 200 µm. (**B**) Quantification of Sirius Red-positive area; (**C**) the levels of Col1a1 mRNA, and (**D**) hydroxyproline content in the liver. The results are presented as mean ± SEM (n = 7–9; ** *p* < 0.01).

**Figure 5 biomolecules-13-01293-f005:**
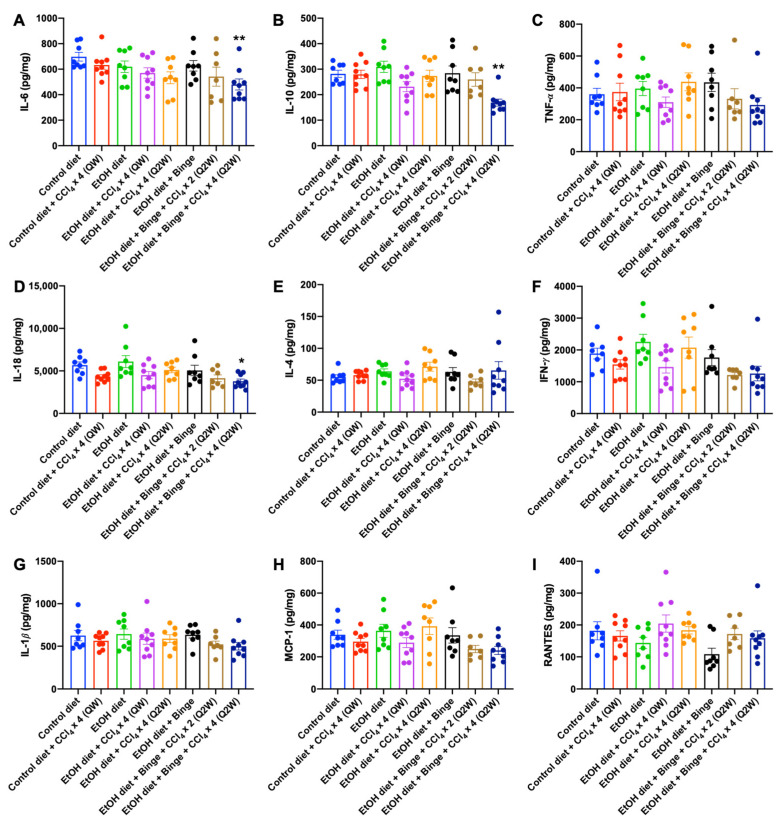
Cytokine and chemokine profiles of animal models. Levels of cytokines (**A**) IL-6, (**B**) IL-10, (**C**) TNF-α, (**D**) IL-18, (**E**) IL-4, (**F**) IFN-γ, and (**G**) IL-1β, and chemokines (**H**) MCP-1, and (**I**) RANTES in the liver. The results are presented as mean ± SEM (n = 7–9; * *p* < 0.05, ** *p* < 0.01).

**Figure 6 biomolecules-13-01293-f006:**
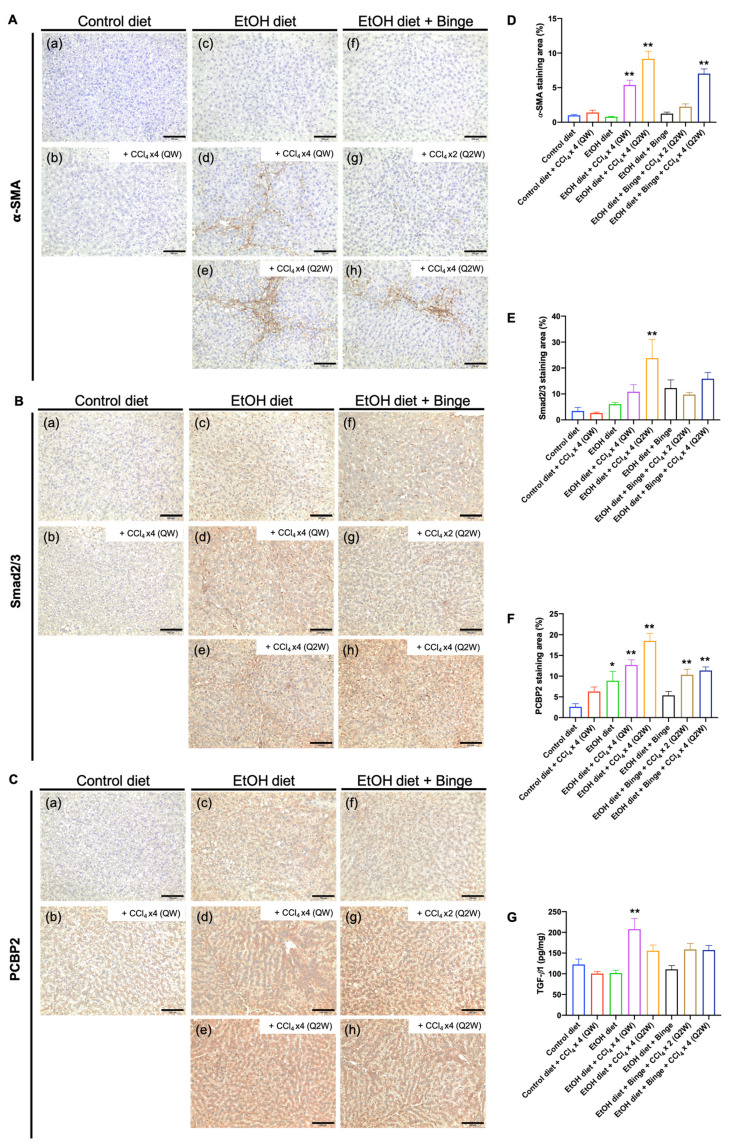
Evaluation of HSC activation in animal models. Immunostaining for (**A**) α-SMA, (**B**) Smad2/3, and (**C**) PCBP2 in liver sections from representative rats treated with (**a**) control liquid diet; (**b**) control liquid diet and four doses of CCl_4_ (QW); (**c**) alcohol liquid diet; (**d**) alcohol liquid diet and four doses of CCl_4_ (QW); (**e**) alcohol liquid diet and four doses of CCl_4_ (Q2W); (**f**) alcohol liquid diet plus binge; (**g**) alcohol liquid diet plus binge and two doses of CCl_4_ (Q2W); (**h**) alcohol liquid diet plus binge and four doses of CCl_4_ (Q2W). The scale bar represents 100 µm. Quantification of (**D**) α-SMA staining area, (**E**) Smad2/3 staining area, and (**F**) PCBP2 staining area. (**G**) Levels of TGF-β1 in the liver. The results are presented as mean ± SEM (n = 7–9; * *p* < 0.05, ** *p* < 0.01).

**Figure 7 biomolecules-13-01293-f007:**
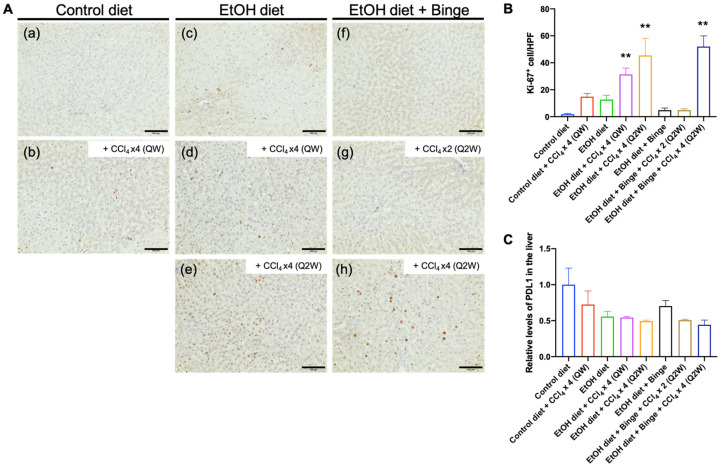
Hepatocyte proliferation and PD-L1 expression in the liver. Immunostaining for (**A**) Ki-67 in liver sections from representative rats treated with (**a**) control liquid diet; (**b**) control liquid diet and four doses of CCl_4_ (QW); (**c**) alcohol liquid diet; (**d**) alcohol liquid diet and four doses of CCl_4_ (QW); (**e**) alcohol liquid diet and four doses of CCl_4_ (Q2W); (**f**) alcohol liquid diet plus binge; (**g**) alcohol liquid diet plus binge and two doses of CCl_4_ (Q2W); (**h**) alcohol liquid diet plus binge and four doses of CCl_4_ (Q2W). The scale bar represents 100 µm. (**B**) The number of Ki67-positive hepatocytes per high power field (HPF), and (**C**) relative levels of PD-L1 expression in the liver. The results are presented as mean ± SEM (n = 7–9; ** *p* < 0.01).

**Table 1 biomolecules-13-01293-t001:** Fibrosis scores of liver specimens.

Model	Liver Fibrosis Models	Fibrosis Scores
a	Control liquid diet	0–1
b	Control liquid diet + CCl_4_ × 4 (QW)	1
c	Ethanol liquid diet	0–1
d	Ethanol liquid diet + CCl_4_ × 4 (QW)	3–4
e	Ethanol liquid diet + CCl_4_ × 4 (Q2W)	3
f	Ethanol liquid diet + Binge	1–2
g	Ethanol liquid diet + Binge + CCl_4_ × 2 (Q2W)	2–3
h	Ethanol liquid diet + Binge + CCl_4_ × 4 (Q2W)	3–4

## Data Availability

The data that support the findings of this study are available from the corresponding author, K.C., upon reasonable request.

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
