# Peer review of "Establishment of a Rat Model of Alcoholic Liver Fibrosis with Simulated Human Drinking Patterns and Low-Dose Chemical Stimulation"

_biomolecules, 2023, doi:10.3390/biom13091293_

Round 1

Reviewer 1 Report

The manuscript by Lin et al describes the Establishment of a Rat Model of Alcoholic Liver Fibrosis with Simulated Human Drinking Patterns and Low-Dose Chemical Stimulation. Experiments are reasonably designed, and results are well presented. Most of the conclusions drawn in this paper are well supported by data, but some aspects of the data need to be clarified and extended:

1. The combination of ethanol and CCL4 to induce alcoholic liver fibrosis has been reported by many other different literatures. For example, CCl4 plus ethanol for 7 wk induces ALD in mice (PMID: 31188634). An ethanol liquid diet containing 4% ethanol plus 2 times IP CCl4 injection per week (8 wk) could also induce liver fibrosis in mice (PMID: 24492981). Therefore, the novelty of the manuscript is weak and should be re-distilled and discussed in the manuscript.

2. Mice (especially, C57Bl6/J) have been identified to be a better and more sensitive model than rats in the study of alcoholic liver disease or liver fibrosis (PMID: 37232739). The rationale to use rats in the current work should be described.

3. AST is one of the well-known and commonly recognized biomarkers for human ALD. However, there were no significant elevations in the plasma levels of AST in any of the groups in the current study (line 224). The mechanisms behind this effect should be addressed and explained properly.

4. The authors stated that rats treated with a combination of alcohol and a low-dose CCl4 were more susceptible to fat accumulation (line 242). To support this claim, Oil Red O staining should be performed.

5. The authors evaluated HSC activation basically through immunostaining (Figure 6). It’s obviously too early to draw the conclusion that the combination of alcohol and CCl4 exhibited a synergistic effect on activation of HSC. To substantiate this assertion, the mRNA and protein levels of α-SMA, Col1a1, MMP2, TIMP-1 or other profibrotic marker proteins should be investigated using RT-qPCR and Western Blots.

6. It would be interesting and more convincing to compare the transcription of fibrosis, proliferation, and inflammation-related genes in animal models by RNA-sequencing with RNA-seq datasets from human liver samples with fibrosis.

7. How about the conditions of hepatocyte apoptosis, hepatic neutrophil infiltration?

Author Response

We sincerely thank the reviewers for their insightful comments to improve the quality of this manuscript. We have accordingly revised the manuscript as per these comments. All revisions in the text have been highlighted in yellow. The following are our point-by-point responses to the reviewers’ comments.

Reviewer 1:

The manuscript by Lin et al describes the Establishment of a Rat Model of Alcoholic Liver Fibrosis with Simulated Human Drinking Patterns and Low-Dose Chemical Stimulation. Experiments are reasonably designed, and results are well presented. Most of the conclusions drawn in this paper are well supported by data, but some aspects of the data need to be clarified and extended:

  1. The combination of ethanol and CCL4 to induce alcoholic liver fibrosis has been reported by many other different literatures. For example, CCl4 plus ethanol for 7 wk induces ALD in mice (PMID: 31188634). An ethanol liquid diet containing 4% ethanol plus 2 times IP CCl4 injection per week (8 wk) could also induce liver fibrosis in mice (PMID: 24492981). Therefore, the novelty of the manuscript is weak and should be re-distilled and discussed in the manuscript.

Thanks for this insight comment. The purpose of this study is to create a simple and feasible method for inducing alcoholic liver fibrosis in a rat model. We also aim to minimize the impact of CCl4 on the initiation of liver fibrosis. Our findings show that 4 i.p. injections of low-dose CCl4 (0.1 mL/kg) effectively induce liver fibrosis in wild type rats fed with an alcohol diet. By contrast, 8-12 i.p. injections of CCl4 (1 mL/kg) are needed to establish liver fibrosis in rats without alcohol feeding.

In the article (PMID: 24492981), a total of 16 injection of CCl4 (0.1 mL/kg) were administered to induce liver fibrosis in mice fed with alcohol. In contrast, 8-12 i.p. injections of CCl4 (0.5 mL/kg) are needed to establish liver fibrosis in mice without alcohol feeding. Therefore, the impact of CCl4 on the development of liver fibrosis is significantly greater in this model compared to our model.

Meanwhile, we investigated the influence of alcohol binges on the progression of liver fibrosis. This study provided additional insights for the development of a practical model for alcoholic liver fibrosis.

Fror the article (PMID: 31188634), CCl4 was administration through inhalation, which is more complicated and less reproducible procedure compared to i.p. injection.

  1. Mice (especially, C57Bl6/J) have been identified to be a better and more sensitive model than rats in the study of alcoholic liver disease or liver fibrosis (PMID: 37232739). The rationale to use rats in the current work should be described.

This is a great point. Our group previously discovered a peptide ligand that targets the insulin-like growth factor 2 receptor (IGF2R), which is overexpressed in activated hepatic stellate cells (HSCs) (PMID: 25955351). The IGF2R peptide ligand showed a high binding affinity for both rat and human HSCs. Subsequently, we developed an IGF2R peptide-modified nanocomplex for the delivery of an antifibrotic siRNA for treating liver fibrosis. The antifibrotic siRNA nanocompelex effectively reversed a CCl4-induced liver fibrosis model in rats (PMID: 33072857). Consequently,  the objective of this study is to establish a feasible and rapid liver fibrosis model in rats, which is induced by alcohol and low-dose CCl4). This model aims to assess the antifibrotic activity of the potential therapeutics.

Furthermore, rats are genetically and physiologically closer to humans than mice. This makes rat a preferred animal model for biomedical research, especially studies related to gene modulaton.

  1. AST is one of the well-known and commonly recognized biomarkers for human ALD. However, there were no significant elevations in the plasma levels of AST in any of the groups in the current study (line 224). The mechanisms behind this effect should be addressed and explained properly.

This is a great question. While both ALT and AST are widely used as a marker for liver injury, they lack specificity for the liver. [PMID: 32407333] Compared to AST, ALT is a relatively specific marker for liver injury because of its higher concentration in the liver than other tissues. By contrast, AST is less specific for the liver because the heart has the highest concentration of AST. Moreover, AST is present in various other tissues including muscles, kidneys, brain, pancreas, and erythrocyte. [PMID: 29479139] We did observe an elevation in ALT levels in the rats treated with alcohol and low-dose CCl4 Figure 2I.

Another potential reason is that ALT and AST levels primarily indicate acute liver injury. Consequently, their levels in case of chronic liver injury are lower than those observed in acute liver injury. For example, Gao et al. observed that in chronic livery induced by chronic alcohol feeding plus multiple binges of alcohol, the serum ALT and AST levels are lower when compared to chronic alcohol consumption combined with a single binge. [PMID: 23449255]

On the other hand, GLDH is widely employed as a specific alternative biomarker for liver injury. This mitochondrial enzyme is mainly located within the liver lobule and is released into the blood from damaged hepatocytes. GLDH has shown better sensitivity and specificity compared to ALT in the detection of liver injuries. [PMID: 32407333, PMID: 23339181

In alignment with this, we observed elevated levels of GLDH in the rats treated with alcohol and low-dose CCl4 Figure 2L.

We have added this discussion in the article.

  1. The authors stated that rats treated with a combination of alcohol and a low-dose CCl4 were more susceptible to fat accumulation (line 242). To support this claim, Oil Red O staining should be performed.

Thanks for this insight comment. Both H&E staining and Oil Red O staining can be employed to observe the damage and fat accumulation in liver cells (PMID: 34943036). In this study, the steatosis of liver in various degrees was very obvious in H&E staining.

  1. The authors evaluated HSC activation basically through immunostaining (Figure 6). It’s obviously too early to draw the conclusion that the combination of alcohol and CCl4 exhibited a synergistic effect on activation of HSC. To substantiate this assertion, the mRNA and protein levels of α-SMA, Col1a1, MMP2, TIMP-1 or other profibrotic marker proteins should be investigated using RT-qPCR and Western Blots.

Thanks for this great suggestion. We presented the expression levels of Col1a1 mRNA in Figure 4C. The results showed that the combination of alcohol and CCl4 significantly increased Col1a1 mRNA expression compared to rats treated with alcohol and CCl4 alone. In addition, the total amount of collagen was measured using hydroxyproline assay.

α-SMA is also a reliable marker of HSC activation (PMID: 18195085). The levels of α-SMA expression were upregulated in the liver tissues in rats treated with alcohol and CCl4, which indicated the activation of HSC.

  1. It would be interesting and more convincing to compare the transcription of fibrosis, proliferation, and inflammation-related genes in animal models by RNA-sequencing with RNA-seq datasets from human liver samples with fibrosis.

This is a great suggestion. We will consider RNA-seqencing in our future studies with this model.

  1. How about the conditions of hepatocyte apoptosis, hepatic neutrophil infiltration?

Our pathologist checked these H&E slides again but did not observe hepatocyte apoptosis and neutrophilic infiltration in alcohol plus low-dose CCl4 models. The inflammatory infiltrates present in the samples were mainly lymphocytes.

Reviewer 2 Report

Dear author, 

I hope you are well and safe while I write this. The study's attempt to develop a rat model of alcoholic liver fibrosis is commendable, considering the challenges in replicating human drinking patterns in experimental rodent models. The introduction of low-dose carbon tetrachloride (CCl4) as an intraperitoneal injection to accelerate liver fibrosis progression is a promising approach. The use of a relatively short 8-week time frame to emulate chronic and heavy drinking patterns is also noteworthy.

The author's findings had a considerable outcome. However, I recommend the author revise the following:

1-    The study should have addressed the potential limitations or drawbacks of this model. For example, the impact of using carbon tetrachloride as an accelerating agent on liver fibrosis should have been discussed, as it can introduce confounding factors and may not perfectly reflect human alcoholic liver fibrosis progression.

2-    In the statistical analysis, the author should indicate the test used for data homogeneity and normality of data.

Yours

Author Response

We sincerely thank the reviewers for their insightful comments to improve the quality of this manuscript. We have accordingly revised the manuscript as per these comments. All revisions in the text have been highlighted in yellow. The following are our point-by-point responses to the reviewers’ comments.

Reviewer 2:

I hope you are well and safe while I write this. The study's attempt to develop a rat model of alcoholic liver fibrosis is commendable, considering the challenges in replicating human drinking patterns in experimental rodent models. The introduction of low-dose carbon tetrachloride (CCl4) as an intraperitoneal injection to accelerate liver fibrosis progression is a promising approach. The use of a relatively short 8-week time frame to emulate chronic and heavy drinking patterns is also noteworthy.

The author's findings had a considerable outcome. However, I recommend the author revise the following:

  • The study should have addressed the potential limitations or drawbacks of this model. For example, the impact of using carbon tetrachloride as an accelerating agent on liver fibrosis should have been discussed, as it can introduce confounding factors and may not perfectly reflect human alcoholic liver fibrosis progression.

Thanks for this insight comment. We have added the description of potential limitations of this model in the disscussion section.

  • In the statistical analysis, the author should indicate the test used for data homogeneity and normality of data.

Thanks for this insight comment. We used ANOVA to analyz quantified data followed by Tukey’s multiple comparison test to evaluate whether there is a significant difference among all groups.

Reviewer 3 Report

The paper entitled "Establishment of a Rat Model of Alcoholic Liver Fibrosis with 2 Simulated Human Drinking Patterns and Low-Dose Chemical 3 Stimulation" developed by Chien-Yu Lin, Evanthia Omoscharka, Yanli Liu, Kun Cheng, attempts to characterize an animal model for the study of alcoholic liver disease in humans, however, in the list of articles cited in this paper and others such as:

-Wang L, Ji G, Zheng PY, Long AH. [Establishment of a rat model of alcoholic liver fibrosis induced by complex factors]. Zhong Xi Yi Jie He Xue Bao. 2006 May;4(3):281-4. Chinese. doi: 10.3736/jcim20060312. PMID: 16696916.

-Yan S, Chen GM, Yu CH, Zhu GF, Li YM, Zheng SS. Expression pattern of matrix metalloproteinases-13 in a rat model of alcoholic liver fibrosis. Hepatobiliary Pancreat Dis Int. 2005 Nov;4(4):569-72. PMID: 16286264.

-Liu HM, Yan M, Zhang XH, Liu L, Shang N, Zhang HT. [Expression of discoidin domain receptor 2 in different phases of alcoholic liver fibrosis in a rat model]. Zhonghua Gan Zang Bing Za Zhi. 2008 Jun;16(6):425-9. Chinese. PMID: 18578992.

-Lamas-Paz A, Hao F, Nelson LJ, Vázquez MT, Canals S, Gómez Del Moral M, Martínez-Naves E, Nevzorova YA, Cubero FJ. Alcoholic liver disease: Utility of animal models. World J Gastroenterol. 2018 Dec 7;24(45):5063-5075. doi: 10.3748/wjg.v24.i45.5063. PMID: 30568384; PMCID: PMC6288648.

They already describe the mechanisms to establish liver damage by chronic alcohol intake with co-administration of CCL4, as well as the hepatic markers of damage such as the study of hepatic stellate cell poducts, so I do not see any novelty in this proposed model, it does not highlight the different characteristics compared to the models already mentioned.

If the differences with the mentioned works are included and highlighted, it can be reviewed again to be accepted for publication.

Author Response

We sincerely thank the reviewers for their insightful comments to improve the quality of this manuscript. We have accordingly revised the manuscript as per these comments. All revisions in the text have been highlighted in yellow. The following are our point-by-point responses to the reviewers’ comments.

Reviewer 3:

The paper entitled "Establishment of a Rat Model of Alcoholic Liver Fibrosis with 2 Simulated Human Drinking Patterns and Low-Dose Chemical 3 Stimulation" developed by Chien-Yu Lin, Evanthia Omoscharka, Yanli Liu, Kun Cheng, attempts to characterize an animal model for the study of alcoholic liver disease in humans, however, in the list of articles cited in this paper and others such as:

-Wang L, Ji G, Zheng PY, Long AH. [Establishment of a rat model of alcoholic liver fibrosis induced by complex factors]. Zhong Xi Yi Jie He Xue Bao. 2006 May;4(3):281-4. Chinese. doi: 10.3736/jcim20060312. PMID: 16696916.

 -Yan S, Chen GM, Yu CH, Zhu GF, Li YM, Zheng SS. Expression pattern of matrix metalloproteinases-13 in a rat model of alcoholic liver fibrosis. Hepatobiliary Pancreat Dis Int. 2005 Nov;4(4):569-72. PMID: 16286264.

 -Liu HM, Yan M, Zhang XH, Liu L, Shang N, Zhang HT. [Expression of discoidin domain receptor 2 in different phases of alcoholic liver fibrosis in a rat model]. Zhonghua Gan Zang Bing Za Zhi. 2008 Jun;16(6):425-9. Chinese. PMID: 18578992.

 -Lamas-Paz A, Hao F, Nelson LJ, Vázquez MT, Canals S, Gómez Del Moral M, Martínez-Naves E, Nevzorova YA, Cubero FJ. Alcoholic liver disease: Utility of animal models. World J Gastroenterol. 2018 Dec 7;24(45):5063-5075. doi: 10.3748/wjg.v24.i45.5063. PMID: 30568384; PMCID: PMC6288648.

They already describe the mechanisms to establish liver damage by chronic alcohol intake with co-administration of CCL4, as well as the hepatic markers of damage such as the study of hepatic stellate cell poducts, so I do not see any novelty in this proposed model, it does not highlight the different characteristics compared to the models already mentioned.

If the differences with the mentioned works are included and highlighted, it can be reviewed again to be accepted for publication.    

Thanks for this insight comment.

Our group previously discovered a peptide ligand that targets the insulin-like growth factor 2 receptor (IGF2R), which is overexpressed in activated hepatic stellate cells (HSCs) (PMID: 25955351). The IGF2R peptide ligand showed a high binding affinity for both rat and human HSCs. Subsequently, we developed an IGF2R peptide-modified nanocomplex for the delivery of an antifibrotic siRNA for treating liver fibrosis. The antifibrotic siRNA nanocompelex effectively reversed a CCl4-induced liver fibrosis model in rats (PMID: 33072857). Consequently,  the objective of this study is to establish a feasible and rapid liver fibrosis model (8 weeks) in rats, which is induced by alcohol and low-dose CCl4). This model aims to assess the antifibrotic activity of the potential therapeutics.

Furthermore, rats are genetically and physiologically closer to humans than mice. This makes rat a preferred animal model for biomedical research, especially studies related to gene modulaton.

For the article 16696916, the rats were treated with a combination of complex factors (alcohol, corn oil, and pyrazole) plus i.p. injection of CCl4 at a dose of 0.3 mL/kg twice a week for 12 weeks. The duration of this model is 4 weeks longer than our model, and the dose of CCl4 is higher than ours. Moreover, pyrazole is used as a inhibitor of alcool oxidation, but it is toxic for both rats and mice. As a result, the impact of alcohol to liver fibrosis in this model is less significant when compared to its effect in our model.

For the article 16286264, the rats were given alcohol (44%, 7g/kg) by oral gavage every day for 24 weeks to develop prominent fibrosis. However, there are notable issues with this approach. First, the daily oral gavage of alcohol is both time-consuming and challenging. Secondly,  employing a model that requires 24 weeks to establish is not practical, particularly when attempting to assess the effectiveness of antifibrotic agents.

For the article 18578992, the rats were give alcohol by oral gavage for 12-20 weeks.

The review article (PMID: 30568384) revealed that rodent models treated with long-term alcohol feeding along with multiple binges and second hit injection showed a high mortality rate. Therefore, we reduced the dose and frequency of CCl4 injections in our co-adminostration model. Meanwhile, we reported that 4 doses of low-dose CCl4 (0.1 mL/kg) ensured the reproducibility of liver fibrosis and accelerate the progression of liver fibrosis within 8 weeks, which is more efficient than the list of articles (at least 12 weeks).

Round 2

Reviewer 1 Report

The authors have addressed my questions.

Reviewer 3 Report

The authors of the paper: Establishment of a Rat Model of Alcoholic Liver Fibrosis with Simulated Human Drinking Patterns and Low-Dose Chemical Stimulation.
They have adequately addressed the points I made to the previous version.